# Role of Stomatal Conductance in Modifying the Dose Response of Stress-Volatile Emissions in Methyl Jasmonate Treated Leaves of Cucumber (*Cucumis Sativa*)

**DOI:** 10.3390/ijms21031018

**Published:** 2020-02-04

**Authors:** Yifan Jiang, Jiayan Ye, Bahtijor Rasulov, Ülo Niinemets

**Affiliations:** 1Institute of Agricultural and Environmental Sciences, Estonian University of Life Sciences, Kreutzwaldi 1, 51006 Tartu, Estonia; qdjyf@hotmail.com (Y.J.); changzhouyjy@hotmail.com (J.Y.); bahtijor@ut.ee (B.R.); 2College of Horticulture, Nanjing Agricultural University, Nanjing 210095, China; 3Estonian Academy of Sciences, Kohtu 6, 10130 Tallinn, Estonia

**Keywords:** stomatal closure, acute emission, LOX compounds, methanol, photosynthesis, MeJA, abiotic stress interaction

## Abstract

Treatment by volatile plant hormone methyl jasmonate (MeJA) leads to release of methanol and volatiles of lipoxygenase pathway (LOX volatiles) in a dose-dependent manner, but how the dose dependence is affected by stomatal openness is poorly known. We studied the rapid (0–60 min after treatment) response of stomatal conductance (*G*_s_), net assimilation rate (*A*), and LOX and methanol emissions to varying MeJA concentrations (0.2–50 mM) in cucumber (*Cucumis sativus*) leaves with partly open stomata and in leaves with reduced *G*_s_ due to drought and darkness. Exposure to MeJA led to initial opening of stomata due to an osmotic shock, followed by MeJA concentration-dependent reduction in *G*_s_, whereas *A* initially decreased, followed by recovery for lower MeJA concentrations and time-dependent decline for higher MeJA concentrations. Methanol and LOX emissions were elicited in a MeJA concentration-dependent manner, whereas the peak methanol emissions (15–20 min after MeJA application) preceded LOX emissions (20–60 min after application). Furthermore, peak methanol emissions occurred earlier in treatments with higher MeJA concentration, while the opposite was observed for LOX emissions. This difference reflected the circumstance where the rise of methanol release partly coincided with MeJA-dependent stomatal opening, while stronger stomatal closure at higher MeJA concentrations progressively delayed peak LOX emissions. We further observed that drought-dependent reduction in *G*_s_ ameliorated MeJA effects on foliage physiological characteristics, underscoring that MeJA primarily penetrates through the stomata. However, despite reduced *G*_s_, dark pretreatment amplified stress-volatile release upon MeJA treatment, suggesting that increased leaf oxidative status due to sudden illumination can potentiate the MeJA response. Taken together, these results collectively demonstrate that the MeJA dose response of volatile emission is controlled by stomata that alter MeJA uptake and volatile release kinetics and by leaf oxidative status in a complex manner.

## 1. Introduction

Stomata play a vital role in controlling the gas exchange between the intercellular air space and ambient air. Numerous studies have demonstrated that guard cells can process multiple, complex environmental signals and respond by opening and closing the stomata thereby altering of water loss and entry of CO_2_ into the leaf [1,2]. Stomatal closure can be triggered by a series of abiotic cues including reductions in light intensity and air humidity, increases in CO_2_ concentration and ozone exposure and drought stress [3,4,5,6,7,8], and biotic cues such as pathogenic microbes [9,10,11,12].

Among different phytohormones, jasmonates represented by jasmonic acid (JA) and its volatile methyl ester (methyl jasmonate, MeJA) are important stimuli to elicit stomatal closure in a wide range of plant species [13,14,15,16,17,18]. Besides producing reactive oxygen species (ROS) as key messengers in mediating stomatal closure similarly to other types of stresses [19,20,21,22], the mechanism leading to the stomatal closure by MeJA has been linked to a typical series of alterations during stomatal closure, including elevation of cytosolic Ca^2+^ concentration and the depolarized plasma membrane, ultimately leading to the efflux of K^+^ from the guard cells, resulting in the loss of turgor and closure of stomata [23,24,25,26]. Stomatal closure by MeJA treatment, in turn, leads to reduced leaf photosynthesis due to limited CO_2_ availability [27,28]. However, herbivory-dependent JAs and exogenously applied MeJA propagate through the leaf and can also impact mesophyll and epidermal cells, potentially directly impacting photosynthetic machinery and altering stomatal openness via mechanisms independent of guard cells [24,25,26,27,28]. Guard cell independent hydropassive stomatal movements and transient photosynthetic suppression have been routinely observed in leaf wounding experiments [29,30]. In the case of MeJA, hydropassive stomatal responses are expected in the very early phases, 1–10 min after MeJA application, but to our knowledge, the initial effects of MeJA have not been examined so far.

Another typical rapid response of the plants to biotic and abiotic stresses is the elicitation of emissions of volatile products of lipoxygenase (LOX) pathway (also called green leaf volatiles) consisting of various C6 aldehydes and alcohols [31,32,33]. To date, stress-dependent bursts of LOX products have been observed in response to both multiple biotic stresses such as herbivore [31,34,35,36] or fungal attacks [33,37,38,39] and abiotic stresses such as heat and frost [40], flooding [41], ozone [42,43], high light intensity [44], and mechanical wounding [44,45]. In addition, methanol emissions have been characteristically observed under biotic and abiotic stresses [30,46,47]. MeJA application leads to major emissions bursts of LOX volatiles and methanol in a MeJA dose-dependent manner [39]. Plant volatile emission patterns change quantitatively and qualitatively in response to MeJA treatments [39,48,49], but it is currently unclear how MeJA-dependent changes in stomatal conductance affect stress-volatile release. In particular, stomatal closure can reduce both gaseous MeJA entry into the leaf and emission of volatiles out of the leaf, thereby temporarily delaying compound emission and dampening the emission peaks [50,51] and potentially blurring the physiological responses. Water-soluble compounds such as LOX volatiles and methanol can be particularly strongly controlled by stomatal conductance due to high temporal storage in leaf liquid phase, and accordingly, slow rise of the compound gas-phase concentration upon stomatal closure [52]. Due to much greater cuticular resistance than stomatal resistance, MeJA entry and release of stress volatiles is assumed to occur almost entirely through the leaf stomata [51,53,54,55]. However, once the cuticle gets damaged, e.g., due to development of MeJA-dependent lesions, the volatile emission through the leaf cuticle can increase, reducing the stomatal controls on volatile release. 

Cucumber (*Cucumis sativa*) has amphistomatous leaves and is highly sensitive to volatile induction by biotic stimuli [56,57] and exogenous MeJA [39]. In the current study, we used *C. sativus* as a model plant and used a range of MeJA concentrations extending from low (0.2 mM) to high (50 mM) to gain an insight into how MeJA-dependent modifications in stomatal conductance affect photosynthesis and kinetics of volatile emission. Although the highest MeJA concentrations used are beyond the typical physiological range, a previous study demonstrated that plant physiological responses were still dose-dependent even at these high MeJA concentrations [39], and thus, the use of these concentrations allows gaining information of the capacity of plant responses. We hypothesized that MeJA application triggers multiphasic stomatal responses, including early opening of stomata followed by closure in a MeJA dose-dependent manner. We suggested that stomatal effects on the MeJA dose dependence of different volatiles depend on timing of volatile emissions. We expected that methanol emissions that are elicited earlier are less affected than LOX volatile emissions that are elicited later during rapid stomatal closure. In addition, experiments with drought and dark pretreatment were conducted to understand how MeJA responses can be interactively modified by other environmental factors that induce stomatal closure. 

## 2. Results

### 2.1. Dose Response of Photosynthesis and Stomatal Conductance to MeJA Treatment 

Before the exposure to MeJA, net assimilation rate (*A*) and stomatal conductance to water vapor (*G*_s_) both increased after enclosure of leaves in the cuvette, and they stabilized after about 30 min of acclimation in the cuvette (Figure 1A,B). After exposure to MeJA, *A* first decreased in MeJA dose-dependent manner followed by a recovery in 20–30 min for MeJA concentrations with 0 (5% ethanol solution) to 5 mM. Nevertheless, *A* remained about 7%–33% lower for these MeJA concentrations even 60 min after MeJA treatment (Figure 1A). For MeJA concentrations 10–50 mM, *A* continuously decreased, and stabilized at -102%–123% of the initial value by the end of the experiment (Figure 1A). Overall, the relative change in *A* was negatively correlated with MeJA concentration (Figure 1C). 

Differently from *A*, *G*_s_ initially increased, whereas the increase tended to be greater for higher MeJA concentrations of 20 and 50 mM than for the rest of the concentrations (Figure 1B). After the initial increase, *G*_s_ stabilized at somewhat higher level for MeJA concentrations ≤2 mM (Control, 0.2, 2 mM; Figure 1B). For leaves treated with higher MeJA concentrations (5 mM, 10 mM, 20 mM, 50 mM), *G*_s_ continuously decreased after the initial increase (Figure 1B). The relative change in *G*_s_ at the end of the experiment was negatively associated with MeJA concentration (Figure 1C).

### 2.2. Dose Response of MeJA-Induced Volatile Emissions

Before the treatment, emissions of volatile isoprenoids (data not shown) and lipoxygenase pathway volatiles (LOX, Figure 2A) and methanol (Figure 2B) were at a low level, close to baseline for most leaves. Emissions of both LOX compounds (Figure 2A) and methanol (Figure 2B) started almost instantly after exposure to MeJA, albeit there was a delay of 5–10 min for the start of the rise of LOX compounds (Figure 2A). The maximum emission rate of total LOX compounds and the integrated emission for the 60 min measurement period following the treatment were strongly correlated with MeJA concentration (Figure 2C). Analogous correlations with MeJA concentration were also observed for the maximum emission rate of methanol and the integrated emission for the 60 min period following the MeJA treatment (Figure 2D). However, the LOX emissions elicited were almost absent for control treatment and for the lowest MeJA concentrations of 0.2 mM, while significant rises of methanol emissions were observed for these two treatments (cf. Figure 2A,B). 

In the case of LOX emissions for MeJA treatments ≥2 mM, the initial rate of emissions was similar, except for the 50 mM treatment (Figure 2A). In addition, the rise of LOX emissions for MeJA treatments ≥2 mM continued longer in treatments with greater MeJA concentrations and thus, the maximum emissions were observed later (e.g., 30–50 min after start of treatment for 50 mM MeJA) than in lower concentrations (e.g., 20–25 min after start of treatment for 2–5 mM MeJA; Figure 2A). 

Differently from LOX, the rate of increase of methanol emissions was greater in treatments with higher MeJA concentration, and the peak methanol emission was typically observed earlier for higher MeJA concentrations, although not always (e.g., 0.2 and 2 mM MeJA treatments in Figure 2B). In general, methanol emissions peaked earlier (10–20 min after the treatment, Figure 2B) than LOX emissions (20–50 min after the treatment, Figure 2A).

### 2.3. Effects of Dark and Drought Pretreatments on the Responses of Photosynthesis and Stomatal Conductance to MeJA Treatment

When the leaf was kept in darkness for 30 min prior to MeJA treatment, *G*_s_ (on average ± SE = 40 ± 6.4 mmol m^−2^ s^−1^) was reduced compared with the control treatment (illuminated plant, 146.5 ± 21.7 mmol m^−2^ s^−1^; Figure 3B). When light was switched on together with 20 mM MeJA treatment, *G*_s_ increased for the first 10 min following MeJA treatment and *A* also became initially slightly positive (Figure 3A,B). However, after the initial rise, both *A* and *G*_s_ declined in dark-pretreated leaves through the 60 min measurement period. The response of the control leaves to MeJA treatment was similar, but the drop in *A* was greater, from 6.4 ± 0.9 µmol m^−2^ s^−1^ before the treatment to −2.9 ± 0.3 at the end of the measurements, ultimately reaching to a lower value than that in leaves kept in darkness (−2.3 ± 0.4) (Figure 3A,C). In addition, the initial increase in *G*_s_ after MeJA treatment was greater in the control treatment than in the dark-pretreated leaves, but 60 min after the treatment *G*_s_ values were similar in the control and dark-pretreated leaves (Figure 3B,D).

Prior to MeJA treatment, the drought-stressed plants (water potential of −1.2 MPa) had a lower *A* (1.9 ± 0.4 µmol m^−2^ s^−1^) and *G*_s_ (84.8 ± 12.6 mmol m^−2^ s^−1^) than the well-watered plants (water potential of −0.2 MPa in well-watered plants). After MeJA treatment, *A* was reduced in the drought-stressed plants, but the reduction was less than that in well-watered plants (Figure 3A,C). However, *G*_s_ of drought-stressed plants was immediately reduced after MeJA treatment and *G*_s_ remained at the low level of 54.04 ± 7.36 mmol m^−2^ s^−1^ until the end of the measurements (Figure 3B,D).

### 2.4. Effects of Dark and Drought Pretreatments on the Responses of Volatile Emissions to MeJA Treatment

Treatment with 20 mM MeJA resulted in induction of both LOX (Figure 4A) and methanol (Figure 4B) emissions in all treatments. The emissions of LOX induced for the dark-pretreated (14.7 ± 2.1 min after MeJA treatment) and for the drought-stressed plants (7.1 ± 0.6 min) were significantly faster (*p* < 0.05) than for the plants in the control treatment (27.6 ± 3.7 min; Figure 4A). Both the maximum and total integrated LOX emissions were the greatest for the dark-pretreated plants, followed by control and drought-stressed plants with significant difference (Figure 4C). 

Peak methanol emissions occurred earlier in the dark-pretreated (15.8 ± 2.1 min) and drought-stressed plants (13.2 ± 2.3 min) than in the plants in the control treatment (21.5 ± 3.8 min; Figure 4B), but without significant difference. The maximum and total integrated methanol emissions during 60 min following the MeJA treatment were similar in dark-pretreated and control plants, but the methanol emission was significantly reduced in the drought-stressed plants (Figure 4B,D).

## 3. Discussion

### 3.1. MeJA Dose-Dependent Changes in Foliage Photosynthetic Characteristics in Cucumber

We observed profound modifications in foliage net assimilation rate (*A*, Figure 1A) and stomatal conductance (*G*_s_, Figure 1B) upon MeJA treatment. In particular, the rapid reduction in *A* and increase in *G*_s_ that occurred within a few min after MeJA application might seem surprising, given that cellular signaling processes triggered by MeJA are more time-consuming. We suggest that this initial rapid change in photosynthetic characteristics is primarily caused by direct osmotic effects of MeJA solution applied in 5% ethanol. A rapid hydropassive stomatal opening is known to occur in response to abrupt reductions in water potential of epidermal and mesophyll cells, so-called Ivanov effect [29]. Such transient stomatal opening has been shown to be induced by multiple factors including leaf cutting, severing and rapid exposure to dry air [29,58,59]. It has been further demonstrated that in conditions of hydropassive stomatal opening, leaf photosynthesis transiently declines, possibly reflecting leakage of solutes from mesophyll cells followed by recovery and stabilization of ionic relations [29]. We suggest that an analogous effect occurs in MeJA-treated leaves as the treatment initially leads to an osmotic shock and associated water efflux from the mesophyll and epidermal cells and hydropassive stomatal opening and reduction in *A* (Figure 1A,B). The responses observed were partly independent of MeJA as *G*_s_ increased and *A* reduced in the control treatment (5% ethanol, i.e., about 1.1 M) as well (Figure 1A,B). A previous study provides overview of rapid decreases of stomatal aperture and is consistent with our proposed hypothesis of an osmotic response [60]. So stomatal aperture is supposed to be observed in our study. However, the time resolution in that study was 30 min, while the responses observed in our study were much faster, and therefore unlikely involving the ABA (abscisic acid) mechanism that was central in that study. Besides the stomatal aperture observation, continuous measurement of stomatal conductance by water vapor exchange is a very well validated method, and although we lost a few mins of data right after application of MeJA, we believe that this method is superior for our purpose under dynamic conditions because of much higher time resolution than 30 min that can be obtained by microscopic techniques that are well suitable for steady-state applications.

Nevertheless, the effect of MeJA was much stronger than the effect of ethanol alone, especially considering the concentration, 1.1 M for ethanol vs. 0.2–50 mM for MeJA, and both the initial changes in *A* and *G*_s_ were quantitatively related to MeJA concentration (Figure 1A,B). In fact, there is evidence that wounding rapidly elicits JA-dependent signaling and that exposure to MeJA leads to rapid wounding-type responses [61,62]. Early wounding responses that can occur within minutes include membrane depolarization, activation of Ca^2+^-dependent signaling and activation of MAP (Mitogen-Activated Protein) kinases [35,62,63,64]. Plasma membrane depolarization in response to elicitors or dehydration is known to lead to rapid reductions in photosynthesis due to alterations in chloroplast structure and ion homeostasis [65,66]. It has been demonstrated that gaseous MeJA taken up by leaves indeed can lead to the elevation of cytosolic Ca^2+^ concentration, depolarized plasma membrane that drives the efflux of K^+^ from the cell, resulting in turgor loss [23,24,25]. Thus, we suggest that MeJA directly elicits rapid changes in photosynthesis characteristics through wounding-dependent signaling, albeit the exact mechanism currently remains unclear.

We further observed that after the initial reduction, *A*, recovered for lower MeJA treatment concentrations of 0–5 mM, while A decreased through the measurement period, and through all MeJA concentrations, there was a quantitative relationship between the relative change in *A* and MeJA concentration (Figure 1A,C). Analogously, after the initial rise, *G*_s_ stabilized in treatments with lower MeJA concentration, while Gs decreased in treatments with MeJA concentration ≥5 mM, and there was a strong correlation between the relative change in *G*_s_ and MeJA Concentration (Figure 1B,C). We argue that these longer-term changes in MeJA-treated leaves primarily result from generation of ROS and elicitation of a programmed cell death (PCD)-like process. There is evidence that MeJA does elicit ROS production in mesophyll cells and that ROS formation precedes photosynthetic dysfunction and PCD [67]. Stress-dependent ROS production can occur in the apoplast, mitochondria and chloroplasts [67,68,69]. As apoplast is the first barrier to gaseous and liquid-phase elicitors, apoplastic ROS release is often elicited first, followed by transfer of apoplastic ROS to ROS and elicitation of ROS burst there [69]. Chloroplasts harbor multiple ROS-producing centers, and chloroplastic ROS production and onset of PCD might also occur independently of apoplastic responses [27,67]. Thus, we suggest that the second slower response to MeJA treatment results from induction of ROS production in chloroplasts, induction of cellular death and subsequent photosynthetic collapse, including decreasing of *G*_s_ and *A* as has been also observed in leaves of *Vitis vinifera* [70,71]. Over a longer term from a few hours to 12–48 h, PCD will results in propagation of lesions and visible leaf damage [39], and decreased expression of photosynthesis-related genes such as ribulose-1,5-bisphosphate carboxylase/oxygenase (Rubisco) and reduced photosynthetic capacity [72,73,74]. We have previously demonstrated that the fraction of leaf area undergoing cellular death is quantitatively related to MeJA concentration, reflecting that MeJA response depends on the number of cells inflicted by stress and the intensity of stress exerted on individual cells [39]. We argue that the MeJA concentration-dependent reduction of photosynthetic activity can be explained by the same mechanism, reflecting positive correlations with the quantities of ROS produced in chloroplasts and the severity of leaf damage. The postulated release of ROS during green leaf volatile (GLV) and methanol emission in our study is supported by previous studies, which use release of stress volatiles as a substitute of ROS formation [75,76,77,78]. Lower resolution studies [79,80] do demonstrate ROS formation simultaneously with emissions of these key volatiles observed in our study. Unfortunately, high-resolution ROS estimation procedure that could accompany the volatile emission is not available. It also suggests that future work is needed to develop methods for high-resolution ROS detection to support our hypothesis.

### 3.2. MeJA Concentration-Dependent Emissions of LOX and Methanol

Time-dependent changes in LOX volatile and methanol emissions broadly supported the postulated impacts of MeJA on changes in cellular oxidative status and propagation of lesions. Both the integrated emission and maximum emission rate of LOX compounds and methanol scaled positively with the concentration of MeJA in our study (Figure 2A–D), indicating that greater MeJA concentrations were associated with more severe oxidative stress. Furthermore, we observed that methanol emissions were elicited before LOX emissions (cf. Figure 2A,B). Stress-dependent methanol release is associated with activation of pectin methylesterases [81], whereas there is recent evidence demonstrating that pectin methylesterase activity is vastly enhanced by H_2_O_2_ [82]. Thus, the faster induction of methanol release suggests that apoplastic ROS formation preceded ROS production in the cytosol.

These quantitative relationships are in agreement with former studies demonstrating stress severity-dependent release of LOX compounds and methanol for a variety of stresses including ozone [46,47,83], heat [40,84], mechanical wounding [30], insect herbivory [35,36,62]. In a previous study, we have demonstrated analogous MeJA concentration-dependent LOX and methanol emission responses [39]. However, in another our study [39], the maximum emission rate of LOX compounds during the first rapid LOX emission burst was about a factor of 2–5 fold (MeJA treatment concentrations ≤20 mM) to 10-fold (50 mM MeJA), and methanol emission about a factor of 1.5–2 fold greater than observed in our study. In addition, in our previous study [39], the time for the peak LOX emission after MeJA application decreased with increasing MeJA concentration and the time to peak methanol emission increased with MeJA concentration, while the opposite was observed in our study (Figure 2A,B). We suggest that these differences among the studies reflect the degree of stomatal opening prior to MeJA treatment. MeJA was applied to leaves with fully opened stomata [39], while in the current study, MeJA was applied at the phase of stomatal opening when the maximum *G*_s_ had not yet been reached (Figure 1B). Thus, we argue that these data are consistent with the hypothesis that MeJA primarily penetrates the leaves through the stomata not through the epidermis [39], and consequently, the total amount of MeJA that entered the leaves at any given MeJA concentration was less in our study than previously reported [39].

Regarding the delay in peak LOX emissions in our study especially for higher MeJA, concentrations, we suggest that this reflects the impact of changes in *G*_s_ after MeJA application on compound diffusion out of the leaves. In particular, oxygenated volatiles such as LOX compounds and methanol are water-soluble, and their emissions can be controlled by changes in stomatal closure [52,55,85]. Once stomata close, volatile gas-phase concentrations gradually increase and ultimately the increase in the diffusion gradient compensates for the reduction in *G*_s_. However, the time required for the increase in the diffusion gradient depends on compound partitioning between gas and liquid phases. For water-soluble compounds, the rise of the diffusion gradient is much slower than for non-water-soluble compounds because a given liquid-phase pool size supports a much lower gas-phase pool size, and accordingly large increases in the aqueous phase concentration are necessary to support increases in gas-phase partial pressure sufficient to re-establish the steady-state emissions [52,55,85]. Therefore, the rate of emission of water-soluble compounds depends on stomatal conductance and the rate of compound production [51,52,55,85]. Thus, it is the time required to achieve a new steady state following a perturbation (e.g., stomatal closure, change in production rate) that determines whether or not the emissions of not soluble BVOC (Biogenic volatile organic compounds) are under stomatal control [51,52,53,55]. In our study, after MeJA application, *G*_s_ transiently increased for a short period of time of 5–10 min during which LOX emission was only slightly induced (cf. Figure 1B and Figure 2A). The rising phase of stomatal opening was followed by stabilization for lower MeJA concentrations and MeJA concentration-dependent reduction in *G*_s_ for greater MeJA application concentrations (Figure 1A). We suggest that increasingly stronger closure of stomata in leaves exposed to greater MeJA concentration is responsible for progressive delay of peak MeJA emission rate in these leaves (Figure 2A).

The situation is different in the case of methanol emissions that were induced more rapidly that LOX emissions (cf. Figure 2A,B). The rise of methanol emissions coincided in large part, between 10–15 min after the MeJA exposure, with the temporal stomatal opening (cf. Figure 1B and Figure 2B). Thus, as the result, methanol release was initially not curbed by stomata, and the methanol release kinetics reflects the rate of methanol formation. However, we argue that the onset of stomatal closure at 20 min after MeJA application, did affect methanol release by reducing the peak methanol emission rate and extending the emission period of methanol release after cessation of its synthesis.

We cannot rule out that some methanol and LOX emissions occurred through the leaf cuticle. Although the cuticular resistance is typically two orders of magnitude higher than stomatal resistance [51,53,54,55], epidermal damage as the result of PCD in MeJA-treated leaves could have strongly reduce cuticular resistance. However, necrotic lesions on leaf surface were visible 3–8 h after the treatment [39], suggesting that changes in cuticular resistance could occur at the later stages of MeJA response. Furthermore, the delayed release of LOX compounds in the treatments with higher MeJA concentration is not consistent with the possible large share of cuticular release of stress volatiles.

### 3.3. Impacts of Dark and Drought Pretreatments on MeJA Responses

Plant stress history and pre-stress physiological status can importantly affect volatile emission rates, composition and time kinetics induced by subsequent stresses [37,48]. In our study, we explored the effects of drought and dark pretreatments on MeJA responses of the net assimilation rate, stomatal conductance, and BVOC emissions. The effects of MeJA on photosynthetic characteristics and volatile emissions were mostly qualitatively similar for control and drought-stressed leaves exposed to MeJA in the light (Figure 3 and Figure 4). However, in drought-stressed plants, the initial rising phase of *G*_s_ was absent (Figure 3B), and changes in *A* and *G*_s_ and induced LOX and methanol emission rates were less than in the other treatments (Figure 3 and Figure 4). This evidence collectively suggests that the amount of MeJA taken up in drought-stressed plants was less than in well-watered plants, and further confirms the suggestion that MeJA enters the leaves primarily through the stomata.

Application of MeJA treatment simultaneously with illumination of predarkened leaves, resulted in surprisingly high LOX and methanol emissions (Figure 4A,B), especially given that *G*_s_ at the time of MeJA exposure was the lowest of all treatments (Figure 3B). Given the low *G*_s_, we suggest that the high LOX compound release not only reflects the MeJA response, but is also directly associated with extra oxidative stress generated by leaf illumination. In particular, photosynthetic induction upon illumination is time-consuming due to low activity of key Calvin cycle enzymes and low pool sizes of Calvin cycle metabolites [86,87,88,89]. Thus, upon sudden exposure to high light, there is a certain amount of excess intercepted light energy that cannot be used in photosynthesis. The excess light energy can lead to highly reactive triplet chlorophyll formation and ultimately to production of ROS, including singlet oxygen [90,91,92]. Excess energy can be dissipated by non-photochemical quenching, but full engagement of safe dissipation of excess excitation energy also requires a certain activation period [93,94,95] during which ROS are characteristically formed [93]. In fact, there is evidence that exposure to high light can lead to LOX compound release [44,96]. We suggest that the enhanced release of LOX and methanol in dark-pretreated leaves (Figure 4) is due to amplification of MeJA-dependent ROS formation by oxidative stress generated by illumination of dark-adapted leaves and also underscores the important role of chloroplasts in MeJA-dependent volatile production. 

We suggest that this can have important implications for plant performance in the field by allowing water conservation and endurance of drought periods. A natural elevation of ambient MeJA concentrations can occur in the immediate vicinity of the attacked leaves and plants, and in whole herbivore-attacked plant stands, and furthermore, airborne MeJA can be carried to neighboring plant stands. In the case of the attacked leaves, MeJA-dependent stomatal closure can somewhat compensate for excess water loss from wounded leaves, but also render the whole plant stand more resistant to drought. Indeed, exposure to MeJA has been shown to alleviate drought stress in rice, soybean, maize and wheat plants [97,98,99,100]. Altogether, these data suggest that the amount of stress volatiles released upon MeJA exposure not only depends on the amount of MeJA that has been taken up by the leaf, but also of the leaf oxidative status. 

## 4. Material and Methods

### 4.1. Plant Growth Conditions

Cucumber (*Cucumis sativus* cv. Libelle F1, Seston Seemned OÜ, Estonia) seeds were sown in 1 L plastic pots filled with a mixture (1:1) of sand and commercial potting soil (Biolan Oy, Finland) and cultivated according to standard practices in a controlled-conditions plant growth room as in our previous studies [39,40]. In short, light intensity of 300–400 μmol m^−2^ s^−1^ at the level of plants was provided for 12 h light period by Philips HPI/T Plus 400 W metal halide lamps. Air temperature was 24 °C at day and 20 °C at night and air humidity was maintained at 60–70%. The plants were watered daily to soil field capacity, and fertilized every three days with commercial NPK (nitrogen, phosphorus and potassium) fertilizer. Approximately 3–4-week-old, 20–30 cm tall plants with four to five fully expanded leaves were used in the experiments. 

### 4.2. Methyl Jasmonate (MeJA) Treatments

The MeJA treatments were applied as previous report [39]. In short, MeJA (Sigma–Aldrich) was dissolved in 5% aqueous ethanol and sprayed uniformly on the upper and lower side of the leaf; this was the application procedure that was associated with minor non-specific effects in control treatments and with most repeatable MeJA effects on plants [39]. 

Before the start of the experiments, the plants were exposed to dim light of 50–100 µmol m^−2^ s^−1^ for about 30 min. The selected leaf was sealed in the gas-exchange cuvette of a portable leaf gas-exchange system described below, and leaf gas exchange and volatile emissions were monitored for 30 min. During this time period, net assimilation rate reached at 70–90% of maximum value (Figure 1A), and stomatal openness at 30–50% of maximum value (Figure 1B). The cuvette was further opened and 10 mL of MeJA solution was sprayed over the upper and lower surface of the selected leaf to obtain a complete and even coating. When the extra MeJA solution had dripped away, the treated leaf was sealed again (within ca. 1 min after the treatment). To obtain MeJA dose response curves, following MeJA concentrations were used: 0 (control, 5% ethanol), 0.2, 2, 5, 10, 20, and 50 mM. Altogether three replicate treatments at each MeJA application concentration were carried out. Experiments with dead leaves and artificial “leaves” were carried out to correct for the effects of free water evaporation on foliage photosynthetic characteristics as explained in Section 4.4.

### 4.3. Interactions of MeJA Treatments with Abiotic Factors

To investigate the interactive effects of volatile emissions elicited by MeJA with abiotic factors, two additional treatments, MeJA application with dark and drought pretreatments, were carried out. The MeJA standard treatment was conducted with leaves acclimated to light (1000 μmol m^−2^ s^−1^) for 30 min such that the stomata were fully opened. In the case of the dark pretreatment, the light inside the gas-exchange system was turned off for 30 min before the exposure to MeJA. Light (1000 μmol m^−2^ s^−1^) was provided again simultaneously with MeJA treatment. For the drought pretreatment, the plant was non-irrigated for 3 days until the water potential decreased to −1.2 MPa. The water potential of the drought-stressed plant was measured by PMS pressure chamber (Model 600D, PMS Instrument Company, OR, USA). In both additional treatments, 20 mM MeJA solution was applied as explained in Section 4.2.

### 4.4. Photosynthesis and Stomatal Conductance Measurements

Gas-exchange characteristics of leaves exposed to different MeJA concentrations were measured with a portable gas-exchange fluorescence system GFS-3000 (Walz, Effeltrich, Germany) equipped with a leaf chamber fluorometer with an 8 cm^2^ cuvette window area. Light was provided by the LED light source of the leaf chamber fluorometer (10% blue and 90% red light) and the humidity was controlled by a built-in GFS-3000 humidifier. The standard conditions for leaf stabilization in the cuvette were: flow rate of 750 µmol s^-1^, leaf temperature of 25 °C, saturating quantum flux density of 1000 µmol m^–2^ s^–1^, and CO_2_ concentration in the cuvette (*C*_a_) of 400 µmol CO_2_ mol air^–1^. After the leaf was enclosed, leaf net assimilation rate (*A*), transpiration rate (*E*), and stomatal conductance (*G*_s_) were recorded continuously, except for a short period of time when the cuvette was opened for MeJA treatment. 

In addition, we conducted additional measurements with dead (dried) leaves and artificial leaves made of Parafilm M (Sigma–Aldrich) to evaluate the time-dependent changes in apparent *G*_s_ caused by physical evaporation from the surface. In these experiments the within chamber vapor pressure deficit (VPD) was maintained the same as in measurements with real leaves). The real stomatal conductance without the evaporation (*G*_s_) was obtained by considering the time-dependent changes in evaporation in dead and artificial leaves. First, the data for dead and artificial leaves were normalized by subtracting the baseline value and dividing by the maximum apparent conductance. The apparent conductance of the dead or artificial leaf at time *t* corrected for the baseline value, *G*_a0,mock_(*t*), is given as:*G*_a0,mock_(*t*) = *G*_a,mock_(*t*) − *G*_a,mock_(*t* = 0)(1)
where *G*_a,mock_(*t* = 0) is the baseline value before the MeJA treatments (a small correction, >1% for all experiments with artificial leaves). Then the relative apparent conductance was calculated:*g*_r,mock_(*t*) = *G*_a0,mock_(*t*)/*G*_a0,mock,max_(2)
where *G*_a0,mock,max_ is the apparent maximum conductance. Then for the leaf measurements, we calculated the change of apparent leaf stomatal conductance at time *t*, *G*_a_(*t*), after the leaf treatment with MeJA,
Δ*_G_*_a_(*t*): Δ*_G_*_a_(*t*) = *G*_a_(*t*) − *G*_s_ (*t* = 0)(3)
where *G*_s_(*t* = 0) is the initial stomatal conductance right before the treatment. The relative change in Δ*_G_*_a_(*t*) was further computed as:*g*_r_(*t*) = Δ*_G_*_a_(*t*)/Δ*_G_*_a,max_(4)
where Δ*_G_*_a,max_ is the maximum value of Δ*_G_*_a_(*t*). The relative change in Δ*_G_*_a_(*t*) corrected for evaporation (i.e., the change due to physiological modifications in stomatal conductance) was found as:*g*_r,c_(*t*) = *g*_r_(*t*) − *g*_r,mock_(*t*)(5)

Ultimately, the evaporation-corrected stomatal conductance after the treatment was calculated as:*G*_s_(*t*) = *G*_s_(*t* = 0) + *g*_r,c_(*t*)Δ*_G_*_a,max_(6)

Comparison of the measurements with the dried leaf and Parafilm (Appendix A), suggested that the measurements with Parafilm were more representative due to significant leaf-to-leaf variability for dried leaves due to absorption of applied solution by some leaves. Thus, we used the Parafilm data to correct for evaporation effects. Separate response curves were used for control (5% ethanol) and MeJA-treated leaves (Appendix A). The measurements with Parafilm and dead leaves demonstrated that the bulk of the evaporation, 90% occurred within 3–5 min (Appendix A). Because of very high sensitivity of *G*_s_ to this initial part dominated by evaporation, we discarded the data for the first 5 min. For the remaining data, the maximum evaporation correction was 7.6–12.2%. 

### 4.5. Calculation of Relative Changes in Photosynthesis and Stomatal Conductance upon MeJA Treatment

The relative changes in A (RA, %) and Gs (RGs, %) after the exposure to MeJA were calculated as: *RA* = 100(*A*t2 − *A*t1)/((*A*t2 + *A*t1)/2)(7)
where *A*_t1_ is the net assimilation rate just before the exposure to MeJA and *A*_t2_ is the assimilation rate after the stabilization, and analogously for *G*_s_: *R*_Gs_ = 100(*G*_s2_ − *G*_s1_)/((*G*_s2_ + *G*_s1_)/2)(8)
where *G*_s1_ is the stomatal conductance before the exposure to MeJA, and *G*_s2_ is the conductance after the stabilization. The time for stabilization was typically 30–40 min since the exposure to MeJA (Figure 1A,B).

### 4.6. Online Monitoring of the Kinetics of VOC (Volatile Organic Compound) Emission with PTR-TOF-MS Combined with Dynamic Head-Space Volatile Collection

A high-resolution proton-transfer reaction time-of-flight mass-spectrometer (PTR-TOF-MS, TOF8000, Ionicon Analytik, Innsbruck, Austria) described in previous studies [39,45] was used to track the volatile release in real time. The PTR-TOF-MS system was connected to the air outlet of the portable gas-exchange system, and after enclosure of the leaf, all measurements were immediately started.

The operation of PTR-TOF-MS, system calibration and compound detection followed the protocol applied in previous study [45]. In short, the drift tube conditions were 2.3 mbar, 600 V and 60 °C, the measurements were carried out continuously over the mass to charge ratio (*m*/*z*) range of 0–316, and data for 31,250 measurements per second were averaged. The raw PTR-TOF-MS data were post-processed with the PTR-MS Viewer 3.0.0.99 (Tofwerk AG, Thun, Switzerland), and relevant *m*/*z* peaks were integrated as explained in previous study [45]. The time resolution used in this study is 10 s (the averaged data were recorded every 10 s). Methanol was detected as the protonated parent ion with *m*/*z* of 33^+^, while the total emission of volatiles produced within the octadecanoid pathway (LOX products) was taken as the sum of individual ion masses (*m*/*z*) of 81.070 [3-hexenal (frag)], *m/z* 99.080 [(*Z*)-3-hexenal + (*E*)-3-hexenal(main)], *m/z* 57.033 [(*E*)-2-hexenal (frag)], *m/z* 83.085 [hexenol + hexanal (frag)] and *m/z* 101.096 [(*Z*)-3-hexenol + (*E*)-3-hexenol + (*E*)-2-hexenol + hexanal(main)] [30,41,45]. Total monoterpene emission was characterized by the parent ion with *m*/*z* of 137^+^ (*m*_137_), and total sesquiterpene emission by the parent ion with *m*/*z* 205^+^ (*m*_205_). The ratio of the fragment ion with *m/z* 81^+^ to *m/z* 137^+^ in non-MeJA-treated leaves was used to separate the share of mass fragment with *m/z* attributable to monoterpenes and hexenals [30]. The emission rates per unit leaf area were calculated according to previous study [101] considering both the incoming air measurements taken frequently during the measurements and empty chamber measurements before plant enclosure (typically only slightly differing from the incoming air concentrations). PTR-MS-TOF measurements were continued for 60 min after MeJA application.

To study the quantitative emission characteristics of LOX compounds and methanol, the emission maxima (*Φ*_M_) and the total volatile emission bursts after (*I*_T_) during a period of 60 min after MeJA treatment were calculated as our previous report [39]. As for the higher MeJA concentrations, LOX, and methanol emissions did not reach to the baseline level during the measurements, total integrated LOX, and methanol emissions were somewhat underestimated. Extrapolating the induced emissions to the baseline suggested that the underestimation was at most 30%.

### 4.7. Statistical Analyses

All experiments were replicated at least three times with different plants and all data shown are averages ± SE. Effects of MeJA dose on foliage photosynthetic and emission traits were studied by linear or non-linear regressions depending on the shape of the response. Effects of abiotic factors combined with MeJA treatment on *A*, *G*_s_, and volatile emission were analyzed by ANOVA followed by Tukey’s test. The analyses were conducted with SAS (Statistical Analysis System) (Version 8.02. SAS Institute, Cary, NC, USA) and all statistical effects were considered significant at *p* < 0.05.

## 5. Conclusions

Our experiments demonstrate that the response to MeJA strongly depends on *G*_s_ that controls MeJA entry into the leaf. This implies that environmental stresses that reduce *G*_s_ are expected to reduce the plant sensitivity to airborne MeJA as was confirmed in our study for drought stress. Furthermore, changes in *G*_s_ alter the dynamics of water-soluble stress-volatile release from the plants, and this can have significant implications for stress propagation and signaling. A model of the relationships between rapid constitutive emission of volatiles and the role of stomata and cuticle by MeJA treatment interacting with light condition was put forwarded (Figure 5). On the other hand, experiments with dark-pretreated and drought leaves demonstrated that MeJA response also depends on leaf oxidative status, implying that understanding multiple stress interactions to MeJA response is complicated. 

## Figures and Tables

**Figure 1 ijms-21-01018-f001:**
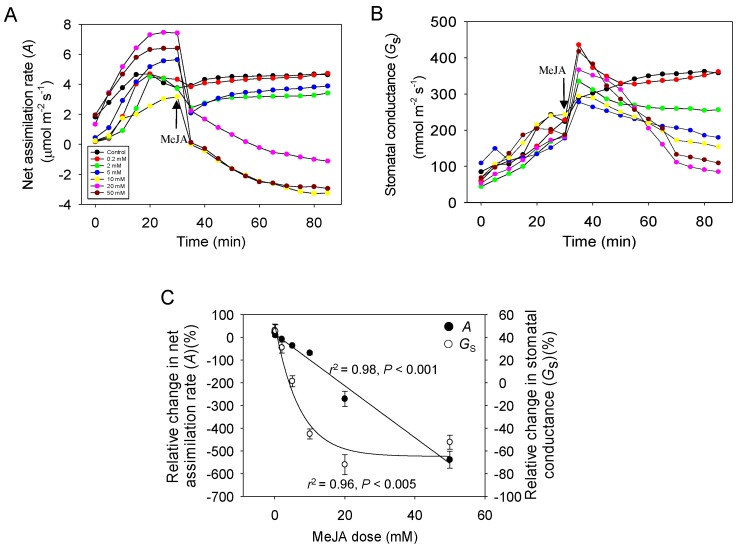
Alteration of net assimilation rate (*A*) (**A**), stomatal conductance to water vapor (*G*_s_) (**B**), and relative changes in *A* and *G*_s_ (Equations (7) and (8)); (**C**) upon exposure to different concentrations of methyl jasmonate (MeJA) in *Cucumis sativus* leaves. All the data are presented on average (*n* = 3). MeJA was dissolved in 5% aqueous ethanol and the concentrations of MeJA applied were 0 (control, 5% ethanol), 0.2 mM, 2 mM, 5 mM, 10 mM, 20 mM, 50 mM. MeJA was applied at the time 30 min after enclosure of leaf in the chamber. The hyperbolic regressions describing the statistical effects of MeJA on the relative change in net assimilation rate and stomatal conductance (**C**), were the following: *y* = 13.9 − 11.4*x*, *r*^2^ = 0.98, *p* < 0.001 (for *A*); *y* = −6.49 + 116.4 × e^−0.14x^, *r*^2^ = 0.96, *p* < 0.005 (for *G*_s_).

**Figure 2 ijms-21-01018-f002:**
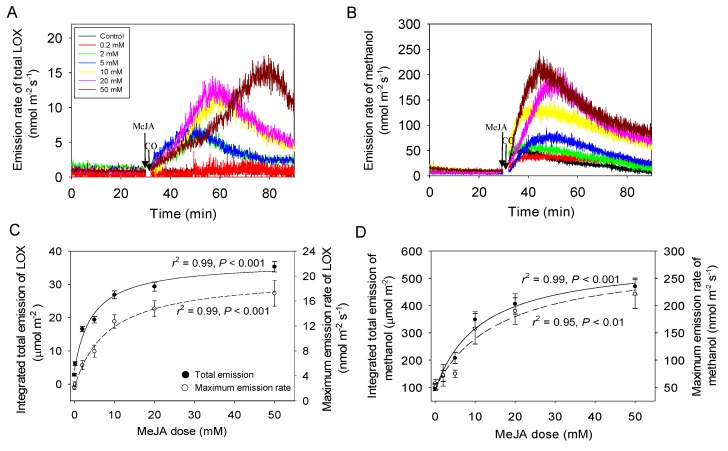
Sample responses of lipoxygenase pathway (LOX) compound (**A**) and methanol (**B**) emissions before (0–30 min) and after (30–90 min) MeJA application and corresponding average ± SE (*n* = 3) total emission and maximum emission rates of LOX compounds (**C**) and methanol (**D**) in *Cucumis sativus* leaves. The applied MeJA concentrations as in Figure 1. The hyperbolic regressions describing the statistical effects of MeJA on LOX (**C**) were: *y* = 5.89 + 31.5*x*/(5.44 + *x*), *r*^2^ = 0.99, *p* < 0.001 (total); *y* = 2.15 + 17.8*x*/(8.34 + *x*), *r*^2^ = 0.99, *p* < 0.001 (maximum). The corresponding regressions for methanol (**D**) were: *y* = 89.8 + 481*x*/(11.4 + *x*), *r*^2^ = 0.99, *p* < 0.001 (total); *y* = 47.91 + 239*x/*(16.2 + *x*), *r*^2^ = 0.95, *p* < 0.01 (maximum). The emission of methanol was calculated from the protonated parent ion with mass/charge (*m*/*z*) ratio of 33^+^ [*m*_33_, methanol]. Total emission of volatiles produced within the octadecanoid pathway (LOX products) was taken as the sum of individual ion masses (*m*/*z*) of 81.070^+^ [3-hexenal (frag)], *m/z* 99.080 [(*Z*)-3-hexenal + (*E*)-3-hexenal(main)], *m/z* 57.033 [(*E*)-2-hexenal (frag)], *m/z* 83.085 [hexenol + hexanal (frag)] and *m/z* 101.096 [(*Z*)-3-hexenol + (*E*)-3-hexenol + (*E*)-2-hexenol + hexanal(main)]. The moment of MeJA spraying (30 min) is denoted. Break time during cuvette opening (about 2 min) is denoted by CO.

**Figure 3 ijms-21-01018-f003:**
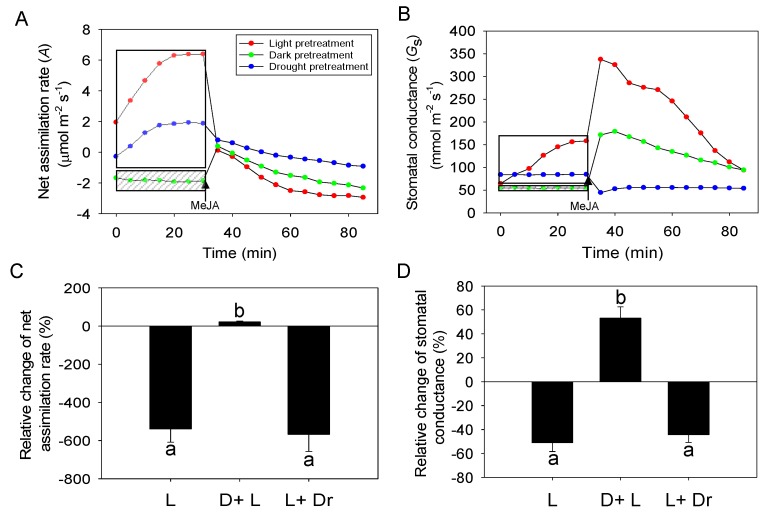
Modifications in net assimilation rate (**A**) and stomatal conductance (**B**) for representative leaves and average ± SE (*n* = 3) relative changes of net assimilation rate (**C**) and stomatal conductance (Equations (7) and (8)); (**D**) upon application of 20 mM MeJA in *Cucumis sativus* subjected to three different pretreatments. All the data are presented on average (*n* = 3). In the light pretreatment (standard treatment applied in all other experiments, the leaf was illuminated 30 min before the treatment. In the dark pretreatment, the leaf was maintained in the darkness for 30 min prior to treatment. In the drought treatment, the plant was non-irrigated for 3 days prior to treatment until leaf water potential decreased to −1.2 MPa. The MeJA application was always conducted in the light and the leaf was illuminated through the rest of the experiment (between 30–90 min). The shaded frame in the figure denotes the dark pretreatment, and the non-shaded frame the pretreatment under light condition. The moment of MeJA spraying (30 min) is denoted. Different letters (a, b) in (**C**) and (**D**) denote statistically significant differences among the means according to ANOVA followed by Turkey test (*p* < 0.05). L: light; D + L: Dark + Light; L + Dr: Light + Drought.

**Figure 4 ijms-21-01018-f004:**
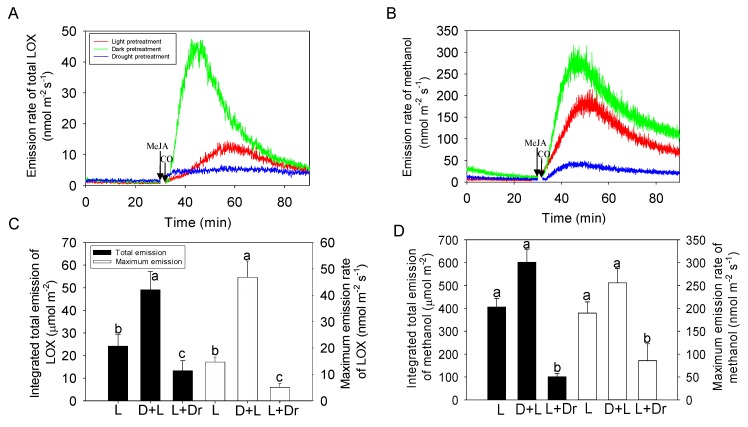
Sample emission time-courses of LOX compounds (**A**) and methanol (**B**) and corresponding average ± SE (*n* = 3) total emission and maximum emission rates of LOX compounds (**C**) and methanol (**D**) in *Cucumis sativus* leaves following exposure to 20 mM MeJA at time 30 min. The leaves were subjected to three pretreatments, leaves illuminated before treatment (light pretreatment), leaves darkened for 30 min prior to treatment (dark) and leaves drought-stressed prior to treatment (drought; Figure 3 for details). Different letters (a, b, c) denote statistically significant differences at *p* < 0.05 (ANOVA followed by Tukey test). The moment of MeJA spraying (30 min) is denoted. Break time during cuvette opening (about 2 min) is denoted by CO. L: light; D + L: Dark + Light; L + Dr: Light + Drought.

**Figure 5 ijms-21-01018-f005:**
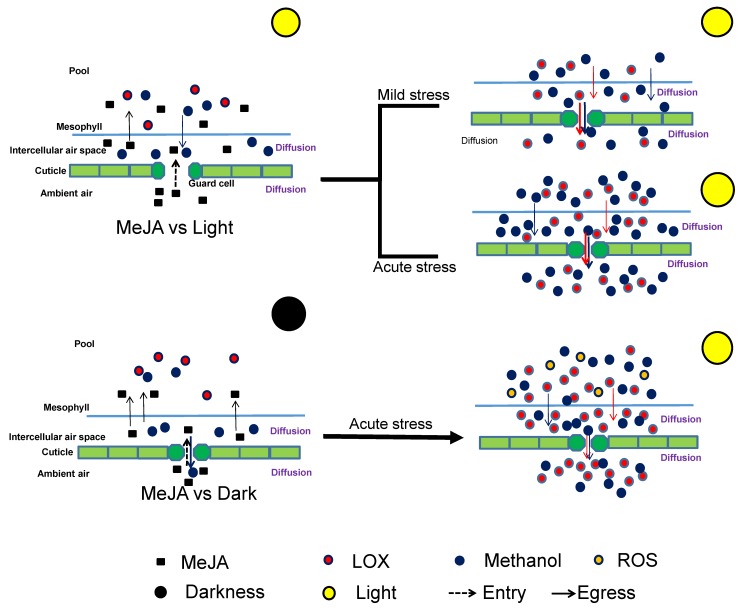
Schematic model of the relationships between rapid constitutive emission of volatiles and the role of stomata and cuticle by MeJA treatment interacting with light condition.

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
