# Peer review of "Role of Stomatal Conductance in Modifying the Dose Response of Stress-Volatile Emissions in Methyl Jasmonate Treated Leaves of Cucumber (Cucumis Sativa)"

_ijms, 2020, doi:10.3390/ijms21031018_

Round 1
Reviewer 1 Report
This paper is very interesting and well designed experiment. However, I do not understand why the authors used such high concentration of MeJA for experiment which may be not reflect real MeJA concentration in plant (cucumber). The author should explained and related with real MeJA concentration in cucumber and this high MeJA concentration in this manuscript.
Overall, the manuscript is very well written.
Author Response
Review 1
This paper is very interesting and well designed experiment. However, I do not understand why the authors used such high concentration of MeJA for experiment which may be not reflect real MeJA concentration in plant (cucumber). The author should explained and related with real MeJA concentration in cucumber and this high MeJA concentration in this manuscript.
Overall, the manuscript is very well written.
Response: We agree that the highest concentration of MeJA in our study is higher than typically observed in cucumber. However, the aim of our study was to investigate the quantitative relation of the MeJA dose with the photosynthetic traits and volatile emission, i.e. confirm the proof-of-concept. So we think the involving the lower concentration (eg. 0.2 mM) and higher concentration (eg. 10, 20, and 50 mM) is necessary. Moreover, a previous study demonstrated that plant physiological responses were still dose-dependent even at these high MeJA concentrations (Jiang et al. 2017), and thus, the use of these concentrations allows gaining information of the capacity of plant responses. We have addressed these points in the MS.
Jiang, Y.F.; Ye, J.Y.; Li, S.; Niinemets, Ü. Methyl jasmonate-induced emission of biogenic volatiles is biphasic in cucumber (Cucumis sativus): a high-resolution analysis of dose dependence. J.Exp.Bot. 2017, 68, 4679–4694.
Reviewer 2 Report
The authors did a good-quality work that shed some light on MeJA dose-response of methanol and LOX volatiles emission that is controlled by stomata. However, some extra experimental work to reinforce the presented data should be performed. First, since the authors state that ''longer-term changes in MeJA-treated leaves primarily result from generation of reactive oxygen species (ROS)'', ROS species detection should be performed in their experimental model. Moreover, I would test the stomatal opening since it is a crucial key point of the whole manuscript. For other comments please check the pdf in attachment.

Author Response
Review 2
The authors did a good-quality work that shed some light on MeJA dose-response of methanol and LOX volatiles emission that is controlled by stomata. However, some extra experimental work to reinforce the presented data should be performed. First, since the authors state that ''longer-term changes in MeJA-treated leaves primarily result from generation of reactive oxygen species (ROS)'', ROS species detection should be performed in their experimental model. Moreover, I would test the stomatal opening since it is a crucial key point of the whole manuscript. For other comments please check the pdf in attachment.
Response: Unfortunately, high-resolution ROS estimation procedure that could accompany these measurements is not available, and studies often use release of stress volatiles as a substitute of ROS formation. Lower resolution studies do demonstrate ROS formation simultaneously with emissions of these key volatiles observed in our study. We have provided relevant references and this statement in the MS.
Line 48 I wouldn't define JAs as biotic stimuli.
Response: “biotic” was deleted.
Line 51-53 this is a general phenomenon, observed not only in presence of MeJA.
Response: Indeed. We have revised the sentence to make this clear.
Line 57 opening
Response: ‘openning’ was changed to opening.
Line 57 reference
Response: Reference was added as suggested.
Line 97 do you mean incubation?
Moreover, what is the rationale behind the choice of those MeJA concentrations? why to not start with the range (0.05-0.1 mM), usually used in plant treatment?
Response: Yes, acclimation in our study mean the incubation of the plants in the cuvette to be adapted to the environment so that the stomata can be full opened.
Line 101 what is this? along the text there are others xx values, please keep attention on that.
Response: Sorry for the mistake. We have replaced the all ‘xx’ with the data.
Line 152 do you mean mM?
Response: Yes. It should be ‘mM’.
Line 169 letters of statistical analysis in panels C and D should be placed more appropriately. Same for other figures along the text.
Response: The place of the letters has been appropriately adjusted.
Line 210 this section is too long, so please make it shorter, toning down statements not corroborated by experimental data.
Response: We have extensively shortened the Discussion part.
Line 247 There are several techniques to measure ROS (see for example Kristiansen, K.A.; Jensen, P.E.; Møller, I.M.; Schulz, A. Monitoring reactive oxygen species formation and localisation in living cells by use of the fluorescent probe CM-H(2)DCFDA and confocal laser microscopy. Physiol. Plant. 2009, 136, 369–383;
Proietti S., Falconieri G.S., Bertini L., Baccelli I., Paccosi E., Belardo A., Timperio A.M., Caruso C. (2019). GLYI4 Plays A Role in Methylglyoxal Detoxification and Jasmonate-Mediated Stress Responses in Arabidopsis thaliana. Biomolecules, 9, 635.
Response: We have added these references. We note, however, that the technique is destructive and cannot applied simultaneously with volatile measurements. Also, the time-resolution is unfortunately not high enough. We hope that with the additional references unequivocally demonstrating the connection between ROS and key volatiles studied here, we hope that we have addressed the limitation that ROS at the required resolution cannot be currently measured.
Line 288 stomatal opening can be monitored by different techniques (for example see: Bak, G.; Lee, E.J.; Lee, Y.; Kato, M.; Segami, S.; Sze, H.; Maeshima, M.; Hwang, J.U.; Lee, Y. Rapid structural changes and acidification of guard cell vacuoles during stomatal closure require phosphatidylinositol 3,5-bisphosphate. Plant Cell. 2013, 25, 2202–2216).
Response: We have included this reference. This study provides a nice overview of rapid decreases of stomatal aperture and is consistent with our proposed hypothesis of an osmotic response. However, the time resolution in that study was 30 min, while the responses observed in our study were much faster, and therefore unlikely involving the ABA mechanism that was central in that study. Continuous measurement of stomatal conductance by water vapor exchange is a very well validated method, and although we lost a few min. of data right after application of MeJA, we believe that this method is superior for our purpose under dynamic conditions because of much higher time-resolution than 30 min that can be obtained by microscopic techniques that are well suitable for steady-state applications.

Round 2
Reviewer 2 Report
The authors addressed all the raised comments in a satisfactory way.